# Molecular Mechanisms of Changes in Homeostasis of the Dermal Extracellular Matrix: Both Involutional and Mediated by Ultraviolet Radiation

**DOI:** 10.3390/ijms23126655

**Published:** 2022-06-15

**Authors:** Alla Zorina, Vadim Zorin, Dmitry Kudlay, Pavel Kopnin

**Affiliations:** 1Human Stem Cells Institute, 119333 Moscow, Russia; doc_zorin@mail.ru (A.Z.); zorin@hsci.ru (V.Z.); 2Department of Pharmacology, Institute of Pharmacy, I.M. Sechenov First Moscow State Medical University (Sechenov University), 119991 Moscow, Russia; d624254@gmail.com; 3N.N. Blokhin National Medical Research Oncology Center, Ministry of Health of Russia, 115478 Moscow, Russia

**Keywords:** skin, extracellular matrix, collagen, aging, fibroblasts

## Abstract

Skin aging is a multi-factorial process that affects nearly every aspect of skin biology and function. With age, an impairment of structures, quality characteristics, and functions of the dermal extracellular matrix (ECM) occurs in the skin, which leads to disrupted functioning of dermal fibroblasts (DFs), the main cells supporting morphofunctional organization of the skin. The DF functioning directly depends on the state of the surrounding collagen matrix (CM). The intact collagen matrix ensures proper adhesion and mechanical tension in DFs, which allows these cells to maintain collagen homeostasis while ECM correctly regulates cellular processes. When the integrity of CM is destroyed, mechanotransduction is disrupted, which is accompanied by impairment of DF functioning and destruction of collagen homeostasis, thereby contributing to the progression of aging processes in skin tissues. This article considers in detail the processes of skin aging and associated changes in the skin layers, as well as the mechanisms of these processes at the molecular level.

## 1. Aging of Human Skin

Skin aging is a complex multifactorial process that develops throughout ontogenesis and affects all biological aspects of this human organ, a process that is determined by both internal factors (endogenous, genetic, and epigenetic) and external factors (exogenous). Among the external factors, ultraviolet irradiation (UV) plays a special role [1,2,3,4,5,6,7]. Thus, there are two types of skin aging: chronological (internal) and photoaging (external). Each type of skin aging has its own clinical and morphological features. The characteristic signs of chronological aging are skin thinning, dryness, decreased elasticity and firmness, paleness, and the presence of fine surface wrinkles. Photoaging occurs under chronic UV exposure; in particular, long-wavelength UVA of 320–400 nm [7] can be observed even before the manifestations of chronological aging. Photoaging results in skin thickening, coarsening, dehydration, deep wrinkles, telangiectasias, irregular pigmentation, and solar lentigo [8,9].

Unlike chronological skin aging which is the genetically determined process depending on a person′s age, photoaging directly depends on the degree of UV exposure, is genetically predetermined by a degree of skin pigmentation [10,11,12], and is considered as an accelerated process of chronological aging [4,13]. The skin in the exposed body areas (face, neck, and hands) undergoes the impact of solar radiation and other environmental factors that cause destructive processes which are believed to be superimposed on chronological skin aging and thereby accelerate its development [14].

Both internal and external aging is primarily associated with phenotypic changes in cells, while the functional manifestations of skin aging are caused by structural and compositional remodeling of the main structural proteins of the extracellular matrix (ECM) in the skin [10].

Skin involution is based on processes occurring at the molecular level which destroy the integrity and structural organization of the genetic material, lead to modification and aggregation of intracellular and extracellular proteins, damage membranes, and cause mitochondrial dysfunction [15]. These processes are accompanied by the decrease in biosynthetic cellular activity, the reduction in the number of active progenitor cells, and the change in molecular composition of ECM. The latter, in turn, is caused by impaired gene expression and post-translational modifications which are mainly distinct in fibrillar proteins. The imbalance occurs between ECM synthesis and decay, which leads to the decrease in tissue mass, the accumulation of degraded molecules in the ECM, and lowering of the functional efficiency of all skin components [5,16].

Aging of each of the skin layers is characterized by its own specific features, which is clinically manifested by thinning of the epidermis, flattening and weakening of the dermo-epidermal junction (DEJ), and the decrease in thickness of the dermis and the hypodermis (Figure 1) [3,17,18,19,20].

## 2. Aging of the Epidermis and the Dermo-Epidermal Junction

Thinning of the epidermis (by 10–50% between the ages of 30 and 80) occurs mainly due to migration of basal keratinocytes (BKs) which have lost their proliferative potential into the spinous layer [21,22,23]. The BK cell population is characterized by the decrease in cell proliferation, quantity, and ability to self-renew. This leads to impairment of epidermal morphogenesis, the decrease in the rate of tissue renewal, and its thinning [24,25,26,27]. Histological examination of skin biopsy samples of elderly people confirms distinct changes in the basal layer of the epidermis, including significant heterogeneity in the size of keratinocytes [28].

One of the main factors contributing to disruption of homeostasis in the BK population is the decreased level of type XVII collagen (COL17), a transmembrane protein of hemidesmosomes which through to attach keratinocytes to the basement membrane (BM) [29]. Apparently, this protein plays a key role in the proliferation of BKs [30]. It has been shown that the resulting differential expression of the *COL17A1* gene in BKs promotes competition among these cells, whereas the protein loss contributes to impairment of homeostasis in the BK pool followed by its depletion, which inevitably leads to the degradation of hemidesmosomes and disruption of epidermal morphogenesis [29].

With age, changes also occur in the population of melanocytes: their functional activity reduces, their number decreases (by 20% every 10 years), and their heterogeneity increases. As a result, uneven skin pigmentation appears [2,26,31,32,33]. There is also a decrease in the activity and number of Langerhans cells (antigen-presenting cells), which causes a decrease in the immune functions of the skin [10,31,34]. As we age, not only a thinning of the epidermis occurs. The production of epidermal lipids is impaired, while the significant decrease in their amount leads to the development of dry skin despite the fact that thickness of the stratum corneum and its hydration remains virtually unchanged [35]. Dryness of the skin is also promoted by the decreased level of hyaluronic acid (HA) in the epidermis and the glycosaminoglycans (GAGs) which have the largest molecular weight and facilitate water binding and tissue hydration (Table 1) [36,37,38].

Age-related destructive changes also occur in the underlying dermal structures, in the dermo-epidermal junction (DEJ) (Figure 1), such as a flattening and reduction in the total surface area of DEJ; a smoothing of the epidermal ridges (due to a decrease in their number and size); a thinning of BM and its multilayering (which apparently has a compensatory character due to flattening of the epidermal ridges [39], while a decrease in the amount of type IV collagen begins after 35 years [40]); as well as the disorganization and degradation of anchoring fibrils [18,19,20].

According to A. Langton (2016), the significant decrease occurs in the level of DEJ main proteins, namely, collagens (types IV, VII, and XVII), integrin β4, and laminin-332, which leads to disturbance of the structural integrity of DEJs and weakens the connection between the epidermis and the dermis [20]. The latter, in turn, causes disruption of metabolic processes between these skin layers and, thus, reduces the supply of nutrients and signaling cytokines to the epidermis, which has a negative effect on the proliferation of BKs by disrupting the homeostasis of their population [21,41,42]. At the same time, the epidermis is also influenced by the state of the dermis, i.e., the water content in “the ground substance” (integrative buffer gel) and the integrity of the collagen–elastin matrix [43].

## 3. Aging of the Dermis

According to the results of histological and ultrastructural studies, the dermis is the skin layer focusing the most significant changes associated with skin aging [13,44]. This occurs primarily due to structural changes in dermal ECM, its reduction, and degradation [2].

The fibers (collagen and elastin) that constitute the skin framework and are the components of “the ground substance”, as well as proteoglycans (PG) and GAGs which play a key role in maintaining firmness and hydration of the skin, undergo particularly prominent degenerative changes [4,10]. As a result of these processes, firmness and elasticity of the skin is lost, its thickness decreases, and wrinkles form [18,34,40,41].

The changes affect both layers of the dermis; the papillary is the upper and thinner layer, while the reticular is the next more pronounced layer [4,10,42,45]. In the papillary layer that is adjacent to BM, a decrease in the content of perlecan (the main PG of BM) and GAG hyaluronic acid is detected (echographically visualized as a subepidermal anechogenic zone [46]), as well as a decrease in the density and spatial orientation of collagen fibers [47,48,49]. In the elastin network of the papillary layer of the dermis, the progressive atrophy of oxytalan fibers is observed, up to their complete disappearance [10,50].

In the reticular layer, a decrease in the density of collagen fibers is accompanied by a decrease in the thickness of their bundles and an increase in the space between them [51], while the thickening of fibers (elaunin and elastin) and a decrease in the number of functional fibers is observed in the elastic network [10,50].

Under the chronic UVA exposure, solar elastosis affects the elastin network of both dermal layers, i.e., there is the deposition and accumulation of elastin masses possessing the incomplete molecular organization and, therefore, the incomplete function. This phenomenon is explained by the stimulating effect of UV on the expression of the gene responsible for the synthesis of elastin. Solar elastosis zones are also characterized by the accumulation of PGs [5,10].

It should be emphasized that changes in the organization and structure of the collagen matrix are characteristic of both chrono- and photoaging of human skin [2]. The results of the study of skin biopsy samples of elderly people have shown that the accumulation of degraded/fragmented fibers and a decrease in de novo collagen synthesis correlate both with age and with the degree of photo-damage severity [51,52,53,54].

Along with degradation of the collagen–elastin matrix, changes also occur in “the ground substance” of the dermis, which is associated with the quantitative and qualitative transformation of GAGs and PGs (Table 1) responsible for hydration and elasticity of the skin. In particular, the bioavailability of HA decreases significantly with age (although its amount remains unchanged [34]), as well as biglycan [36].

Proteoglycan decorin, that is, the smallest in size and the most important regulator of the assembly of collagen fibers, also undergoes quantitative and qualitative changes, while a decrease in the molecular weight of its polysaccharide chains has a significant negative effect on skin elasticity since decorin is involved in fibrillogenesis and determines the diameter of fibrils [36,55]. It has been shown that during photoaging, the abnormal accumulation of HA, versican, and chondroitin sulfate is observed in the solar elastosis zones, while decorin is completely lost; all these phenomena are caused by chronic damage of the skin under UV exposure [36].

Histological examination of skin biopsy samples of young and elderly people confirms the changes described above. Thus, the histological picture of skin sections [4] showed that the “young” and photoprotected skin is characterized by the pronounced epidermal ridges and DEJs, as well as the highly organized network of collagen fibers and a cascade of elastic network fibers connecting BM with the papillary and reticular layers of the dermis; while the chronologically aged skin is characterized by the reduced collagen fibers and elastic network fibers (especially oxytalan fibers) and the reduced content of GAGs, whereas the photoaged skin is characterized by the reduction in collagen fibers, including type VII collagen in the area of DEJs, and by solar elastosis, that is, the accumulation of disorganized proteins of elastin fibers throughout the dermis and also the accumulation of GAGs. In the case of chronological aging of the photoprotected skin, the flattening of DEJs is observed, as well as the disorganization of the elastic network (mostly in the papillary layer), which is accompanied by the accumulation of amorphous elastin and a reduction in the number of collagen fibers. During photoaging, the skin of both young and elderly patients is characterized by the pronounced destruction of epidermal ridges and DEJs, degradation of the elastic network, and accumulation of amorphous elastin in both layers of the dermis. Aging and UV exposure cause the disorders observed in the integrative buffer system of the dermis, a decrease in functioning of the highly organized network of elastic fibers connecting all layers of the skin through cascading, and structural deformations of collagen fibers including their progressive fragmentation. All these events change the essential functional properties of the skin by reducing the skin hydration, elasticity, firmness, and strength [2,4,46,54]. As regards the collagen fragmentation, it is important to note the degradation of type I collagen fibers, the most common structural fiber-forming protein of the skin, which comprises 80–90% of the total collagen amount, while the other two fiber-forming collagens type III collagen and type V collagen are 8 to 12% and up to 5%, respectively [44]. With age, the level of total collagen in the skin decreases (by approximately 1% throughout the entire adult person’s life [56]), and the level of main collagen types I and III also decreases, especially at the age of over 60 [57]. According to other data, the increase in collagen type III/I ratio occurs with age due to the increased degradation of type I collagen [58].

In the process of skin aging, changes are also observed in the skin blood supply, i.e., worsening microcirculation, microhemodynamics, and rheological properties, while the number of blood vessels decreases. This is especially related to the papillary dermis [59] where the capillary bed decreases by almost a third at the age from 40 to 90 years. In the reticular dermis, the capillary bed remains almost unchanged [42]. According to Helmbold et al. (2006), a key role in the deterioration of the microvascular network in the skin is played by a significant decrease in the density of pericytes (multipotent perivascular cells formed by the differentiation of mesenchymal precursors cells in situ) during formation of the endothelium since pericytes together with smooth muscle cells promote the proliferation and migration of endothelial cells as well as the maintenance of newly formed vessels [60]. Therefore, a decrease in the number of pericytes leads to impairment of remodeling of the bloodstream in the skin.

With age, there are also involutional changes in the fibroblastic differon of the dermis which are associated with the changes in amount of fibroblasts, their biosynthetic activity, and the formation of senescent fibroblasts that have a significant impact on the skin aging process (see review A. Zorina et al. (2022)) [61].

## 4. Aging of the Hypodermis

Age-related changes in the hypodermis (subcutaneous adipose tissue) are mainly associated with the reduction in subcutaneous fat [17], which is caused by the decrease in adipogenesis and lipogenic ability (fat accumulation) of adipocytes [62,63]. The decrease in adipogenesis occurs due to the decrease in differentiation of adipocyte precursor cells (multipotent mesenchymal stromal cells) into mature adipocytes (adipose tissue cells) [64,65]. These destructive processes are based on the increased level of anti-adipogenic factors, in particular CHOP (C/EBP Homologous Protein, one of the transcription factors) and TNF-α (tumor necrosis factor), which inhibit early adipogenesis and activate other anti-adipogenic proteins [66]. They block the activity of C/EBPα and PPARγ, the key transcription factors of adipogenesis. The expression and activation of these proteins are necessary conditions for the induction of MMSC differentiation toward adipogenesis [5,44,46]. Inhibition of PPARγ activity is one of the main factors decreasing lipogenic activity of adipocytes, which occurs under the action of protein sirtuin (SIRT1) whose level increases with age [64,65].

An important cause for the suppression of adipogenesis also lies in the formation of senescent cells in the hypodermis (from the Latin word *senex* meaning “aging”) which accumulate in the tissue and secrete many anti-adipogenic factors, in particular, activin A belonging to the superfamily of transforming growth factors, as well as many pro-inflammatory cytokines, including interleukin-6 (IL6), tumor necrosis factor α (TNFα), and interferon γ (IFNγ) [67,68,69]. The latter initiate the chronic nonspecific inflammation contributing to the destruction of stem/progenitor cell niches and their depletion. The consequence of these processes is a decrease in the number of mature adipocytes and, therefore, reduction in subcutaneous tissue.

## 5. Causes of Destruction of the Extracellular Matrix in the Skin

One of the main factors damaging the integrity of the skin framework is the impaired balance between degradation and de novo formation of collagen and elastin fibers. In the adult skin, elastin has a very low rate of renewal (the half-life of elastin in human tissues is over 70 years) [70,71]. The protein structures of the elastic network arise during embryogenesis, whereas the maximal production of elastin fibers occurs in the perinatal period, then decreases and is practically not observed in adults ([43,50]. Collagen synthesis de novo also occurs relatively rarely, while the average half-life of mature cross-linked collagen in human skin is 15 years [72]. Nevertheless, unlike elastin, collagen synthesis de novo occurs in the dermis, in particular, during the physiological process of wound healing when the skin is damaged. During the initial phase of wound healing, degradation of collagen fibers occurs by means of matrix metalloproteases (MMPs), which promote the migration of immune cells into the dermis. Then, the damaged collagen fibers are replaced by de novo collagen production by dermal fibroblasts [73]. 

MMPs comprise a family of endopeptidases that are responsible for the physiological degradation of various extracellular matrix proteins, while MMP-1 (collagenase) is the main metalloprotease that begins the fragmentation of type I and III collagen fibers; after that, the fibers are cleaved by MMP-3 (stromelysin) and MMP-9 (gelatinase) proteases [74,75].

The main producers of MMPs in the skin are keratinocytes and fibroblasts. The activity of MMPs is regulated by specific endogenous tissue inhibitors of metalloproteinases (TIMPs). It has been revealed that with age, a significant increase in the level of MMPs occurs in the human skin, while no adequate increase in the level of TIMPs is observed. Such an imbalance in the activity of MMPs and TIMPs leads to the unregulated destruction of components of the dermal ECM, which promotes the progressive destruction of the skin framework and escalates the processes of both chrono- and photoaging [44,76,77,78,79,80].

It seems obvious that the destructive processes developed in the skin during aging affect all structures and components of the skin layers with no exceptions. The age-related destructive processes are accompanied by impairment of the blood supply in the skin and disturbance of the structural integrity of the skin collagen–elastin framework and its constituent water gel. However, the most significant age-related processes are associated with the dermis and connected with changes in the dermal cell population, fibroblasts, as well as with changes in the structure and function of ECM.

## 6. The Relationship between the State of Collagen Matrix and Functioning of Dermal Fibroblasts

It has been shown that signals entering the dermis from outside are perceived by the ECM and transmitted to DFs, which, receiving these signals, provide ECM homeostasis [13,51,81]. At the same time, the DF functioning directly depends on the state of the surrounding collagen matrix (CM). The intact CM ensures proper adhesion and mechanical tension in DFs, which allows these cells to provide collagen homeostasis while the ECM fully regulates cellular processes including cell migration, proliferation, differentiation, and apoptosis [82].

When the CM integrity is damaged, which occurs both during chronological aging and photoaging, changes are observed in mechanotransduction (transmission of mechanical signals from ECM to cells), promoting development of the mechanism that disrupts DF functions (Figure 2) [13,53,83,84].

It has been revealed that integrins (heterodimeric transmembrane proteins, specific receptors, primarily, α1β1 and α2β1 to collagen type I [85]) located on the cell surface are able to specifically bind ECM proteins, in particular type I collagen [53]. The adhesion of integrins to ECM proteins contributes to the formation of bonds not only between integrins and the collagen matrix but also with actin (protein of the DF cytoskeleton) since integrins are attached to the CM from the outer surface of the cell membrane and connected to the cytoskeleton from the inner surface of the cell membrane, thereby creating focal adhesion complexes (focal contacts) that ensure the closely related regulatory and mechanical functions of DFs [11,53,86]. The formation of these complexes induces a cascade of intracellular signaling pathways that regulate the DF metabolism including the balance between production of collagens and their degradation by MMPs. Due to the focal contacts, DFs can “spread out” on the CM, which allows intracellular microfilaments to exert mechanical pressure on the matrix. At the same time, the cytoskeleton microfilaments located on the inner surface of the cell membrane and in the cytoplasm are physically linked to integrins and use this coupling to tighten the collagen network [13]. The internal tension of actin–myosin microfilaments (AMF) activates the complex of intracellular microtubules and intermediate filaments, contributing to the formation of pressure from the outside. A balance is created between the external pressure and the internal tension of AMF, which results in a dynamic tension between DFs and CM. This allows DFs to achieve the proper level of stretching which ensures the possibility of perfect functioning, including the synthesis of collagen and other ECM components [85]. When the structural integrity of CM is disrupted, the mechanical tension decreases which leads to the reduction in DF focal adhesion and violates the mechanical resistance of collagen fibers. As a result, the balance between the tension inside DFs and the pressure outside them is disturbed. For this reason, DFs lose their ability to stretch and, therefore, reduce the production of collagen, while the production of MMPs, on the contrary, increases, contributing to even greater disorganization of collagen fibers. Thus, the production of collagen in the elderly (80 years and older) compared with its synthesis in the skin of the young (18–29 years) decreases by about 75% [11,53], while the level of collagen degradation (similar to photoaging) increases by 75% [11]. Moreover, there is a parallel decrease in the content of collagens of types I and III, which constitute the main structural fibers of the dermal ECM [79,87].

Despite the different etiologies, the disorders observed in both types of aging are based on the common fundamental molecular mechanisms. The oxidative stress is believed to be the main trigger of these destructive processes [79,88].

## 7. The Role of Oxidative Stress and Other Factors in Changing Collagen Homeostasis

The oxidative stress (OS) can originate not only under the influence of UV but also the factors not connected with UV light. The formation of reactive oxygen species (ROS) independent of UV exposure regularly occurs in DFs and keratinocytes from molecular O2 during aerobic respiration and as a result of cellular metabolism [89]. The formed ROS accumulate in skin cells and, despite the antioxidant protection that decreases with age, initiate cascades of signaling pathways inducing a disturbance of collagen homeostasis (Figure 3) [50].

In the young skin undamaged by UV, the downstream signaling pathways in DFs are maintained at the low level of activity. This occurs due to protein tyrosine phosphatase (PTP) activity that dephosphorylates receptor tyrosine kinases (RTK, transmembrane proteins) (Figure 3) [50,51], which contributes to low MMP production and proper collagen synthesis, therefore preserving the collagen homeostasis in skin tissues.

With age, ROS formed in high concentrations in DFs overcome antioxidant protection, bind to specific sites in the PTP catalytic center, and cause inactivation of the enzyme. The inactivation of PTP leads to the phosphorylation of RTK and activation of downstream signaling pathways including the family of mitogen-activated protein kinases (MAPK). The activation of MAPK, in turn, initiates the expression of factors c-Fos and c-Jun which combine with each other and form activating protein AP-1 (Activator protein 1, transcription factor). This protein is the main factor inducing collagen homeostasis disorders in the skin and is the central link in the pathogenesis of both types of aging (Figure 4) [11,13,53,90,91].

AP-1(c-Fos, c-Jun) plays a key role increasing the transcription of genes responsible for the synthesis of MMPs (MMP-1, -3, and -9) [51]. MMP-1 initiates the cleavage of collagen fibers in the specific site inside the central triplet helix, while MMP-3 and MMP-9 continue this process [76]. As a result, degradation/disorganization of CM occurs [92]. At the same time, the activation of AP-1 leads to a decrease in the synthesis of procollagen types I and III by blocking the signaling pathways of the transforming growth factor TGF-β that is the main regulator of collagen homeostasis in the skin [11,93,94]. TGF-β controls collagen homeostasis by regulating collagen production and degradation through the Smad signaling pathway (TGF-β/Smad) by means of connective tissue growth factor CCN2 [95]. CCN2 is one of six members of the CCN family of matricellular proteins that regulate a variety of cellular functions (from adhesion, migration, proliferation, differentiation, and apoptosis to angiogenesis, synthesis of ECM proteins, and DF–ECM interactions) [96,97].

In addition, AP-1 reduces the level of TGFBR2 (TGF-beta receptor type 2) on the surface of DFs and, thus, inhibits the phosphorylation of proteins Smad 2/3, while Smad 2/3 and Smad 4, in combination, are important transcription modulators of genes responsible for the synthesis of procollagens. Thus, AP-1 induces Smad7, an inhibitory protein of the canonical TGF-β signaling pathway, which, in turn, blocks the TGF-β1 signaling pathway and suppresses the synthesis of procollagens (Figure 3) [98,99,100]. In addition, matricellular protein CCN1 associated with ECM also suppresses the expression of receptor TRII to TGF-β on the surface of DFs and activates AP-1 (Figure 4) [101]. As a result of an increase in the level of CCN1, collagen synthesis decreases and MMP-1 synthesis increases [102]. The high correlation was observed between the increase in the CCN1 level and the decrease in the synthesis of type I collagen, as well as the increase in the MMP-1 level and the increase in fragmentation of collagen fibers [103]. At the same time, a simultaneous decrease was also recorded in the level of CCN2 that is an essential mediator of TGF-β activity in the process of collagen production [98,99,100].

TGF-β is also involved in regulation of autophagy by increasing the expression of autophagy genes (Beclin1, ATG-5, and ATG-7) [104,105]. The autophagy associated with DFs plays an extremely important role in maintaining tissue homeostasis, especially in the case of UV skin damage, since it neutralizes the compounds formed as a result of the OS effect and participates in the repair of damaged cellular structures [105,106]. The reduction in autophagy in DFs promotes the development of inflammation in tissues due to the activation of inflammasomes (multiprotein oligomeric complexes responsible for the development of inflammatory response), and thus the skin-aging phenotype.

Along with the activation of MMPs and suppression of procollagen synthesis through the MAPK signaling pathway, a number of other mechanisms are activated under the influence of UV, which leads to disruption of collagen homeostasis (Figure 3) [50]. In particular, UV induces the synthesis of tumor necrosis factor TNF-α (pleiotropic cytokine activating the synthesis of MMP-1 and MMP-3). Granzyme B (GzmB) is also activated by UV, especially under chronic exposure. Granzyme B is a serine protease that cleaves the small proteoglycan decorin, the most important regulator of the assembly of collagen fibers. Decorin binds to cross-linked collagen fibers at the cleavage site for MMP-1; thus, the binding protects the collagen fibers from cleavage by proteases [51]. The excessive level of GzmB significantly decreases the amount of decorin; therefore, this weakens the protection of collagen fibers from cleavage by MMPs, while MMP-1 receives open access to CM fragmentation [107].

Another important factor disrupting collagen homeostasis is the accumulation of senescent DFs (senDFs) in the skin with age. The senescence-associated secretory phenotype (SASP) is developed in senDFs, which is characterized by the high levels of MMPs and proinflammatory cytokines (Figure 4) [108]. The proinflammatory cytokines have been shown to activate the transcription factor AP-1 which induces the production of MMPs, thus promoting the degradation of ECM proteins and thereby the progression of skin aging including thinning of the skin and the formation of wrinkles [90,91]. SenDFs are also characterized by a marked reduction in the IGF-1 signaling pathway (insulin-like growth factor), which is accompanied by the disruption of epidermal morphogenesis and, therefore, atrophy of the epidermis [91,92]. IGF-1 is synthesized by DFs and participates in the regulation of proliferation and differentiation of epidermal cells [109].

It should be noted that not only ultrastructural, morphometric, and mechanical changes occur in the CM of the dermis, but also conformational ones. There are changes in the collagen macromolecule that develop due to formation of the so-called AGE products (advanced glycation end products) (Figure 2). AGEs are extremely stable and resistant to the proteolytic action of MMPs and appear as a result of the non-enzymatic reaction between carbohydrates and free amino groups of proteins, i.e., reaction of glycation [110,111]. One of the important factors in the formation of AGEs is oxidative stress [112].

The formation and accumulation of AGEs results in the following events:Increase in the stiffness of the dermal ECM;Damage of collagen in terms of its biochemical properties that are essential for the interactions between the ECM and DFs;Disruption of cellular processes, including adhesion and migration of DFs [49,113,114,115].

It has been also revealed that AGEs directly affect DFs by enhancing the expression of genes responsible for the synthesis of MMPs and inhibiting the expression of genes encoding proteins [116].

Thus, with age, the homeostasis of human skin collagen is being attacked by multiple factors.

## 8. Morphological Changes in Dermal Fibroblasts Observed in the Aging Process of Dermal Collagen Matrix

The numerous structural changes observed in the aging process of the dermal CM [50,113], in turn, cause the changes in morphology and size of DFs. Thus, the flat stretched cells having the pronounced nucleus and extended cytoplasm and contacting with a multitude of intact collagen fibers turn with age into compressed (reduced) dilapidated cells with cytoplasmic remnants. At the same time, the open space surrounding DFs increases (the number of DF contacts with CM decreases by an average of 80%, whereas the area of cell surface is reduced by 75%) [11,50,53]. The secretory profile of DFs also changes; collagen synthesis decreases and the production of MMPs increases, which aggravates the disruption of collagen homeostasis in the skin. Thus, the positive feedback stimulating further degradation of ECM is formed [80]. As a result, a vicious circle arises promoting the further progression of skin aging [50].

Apparently, the multilateral influence of ECM on DFs is so significant that it should be more correct to consider their relationship not as an interaction, but as an absolute interdependence [53].

## 9. Conclusions

The dramatic changes occurring in the dermis with age are associated with DFs, the main cellular component, and also with the dermal ECM produced by DFs. A decrease in the number of DFs [50,117], which occurs primarily due to depletion of the pool of DF precursor cells [21,118,119], leads to a decrease in the regenerative potential of skin tissues [91,120,121,122]. At the same time, the structural and conformational changes developing with age in the dermal ECM impair the functioning of differentiated (mature) DFs which are responsible for the remodeling/renewing of the ECM. All these destructive processes form a vicious circle promoting the constant progression of skin aging.

There is no doubt that the study of molecular mechanisms underlying the abovementioned processes and their understanding is extremely important since this opens the way for finding new approaches to solve the problem of skin aging, as well as aging of the other tissues and the whole body. Thus, taking into account the tremendous role of senDFs in tissue aging, currently many laboratories in the world actively develop treatments, the so-called senolytics, that allow one to selectively remove senCs from tissues, as well as senomorphs that inhibit the destructive effect of SASP [123,124,125,126,127]. In addition, the development of methods/medicines that can reduce the stiffness of the dermal ECM, including the SC niches, is no less topical [128]. It can be expected that such innovative approaches allow one to maintain the quantitative and qualitative level of SC functioning and, therefore, the regenerative potential of tissues, thus preventing the progression of aging processes.

## Figures and Tables

**Figure 1 ijms-23-06655-f001:**
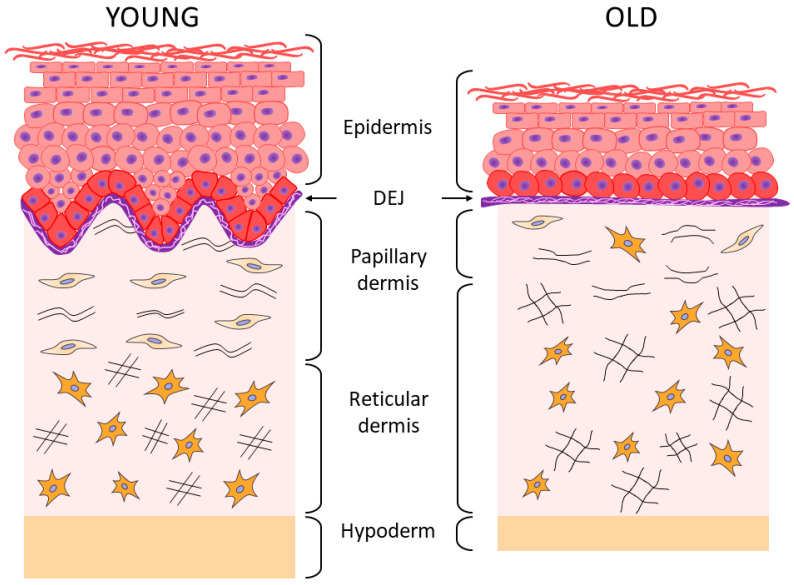
Schematic representation of skin aging.

**Figure 2 ijms-23-06655-f002:**
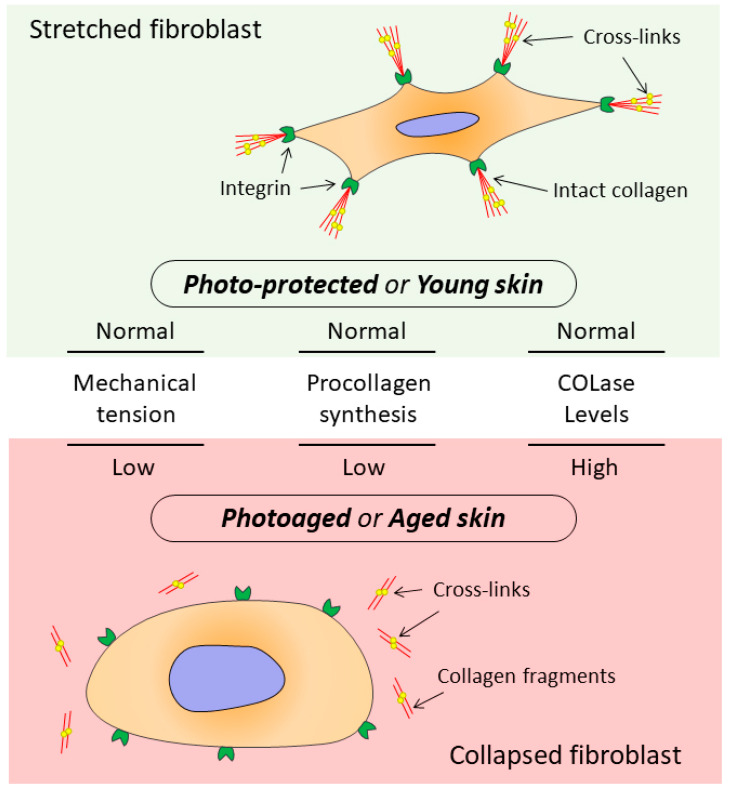
Schematic representation of the relationship between mechanical tension of CM and the DF cytoskeleton during collagen production in the human dermis.

**Figure 3 ijms-23-06655-f003:**
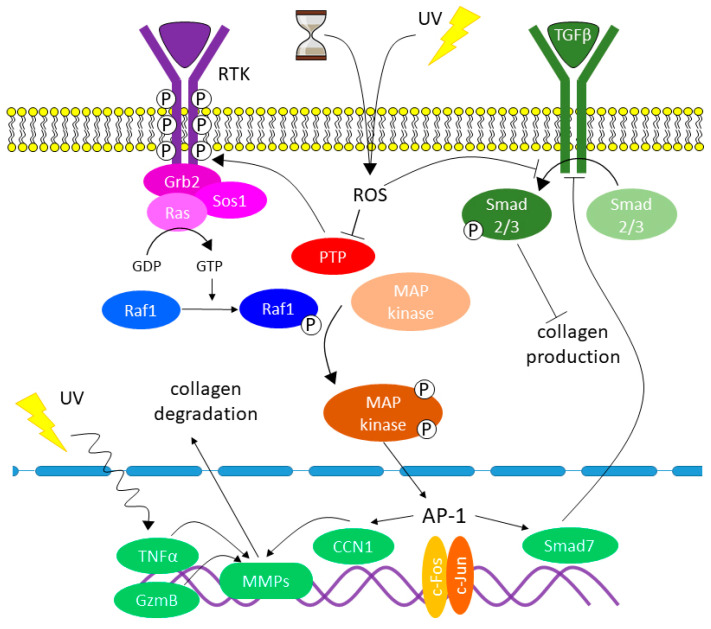
ROS-mediated activation of cascades of molecular signaling pathways in the human skin.

**Figure 4 ijms-23-06655-f004:**
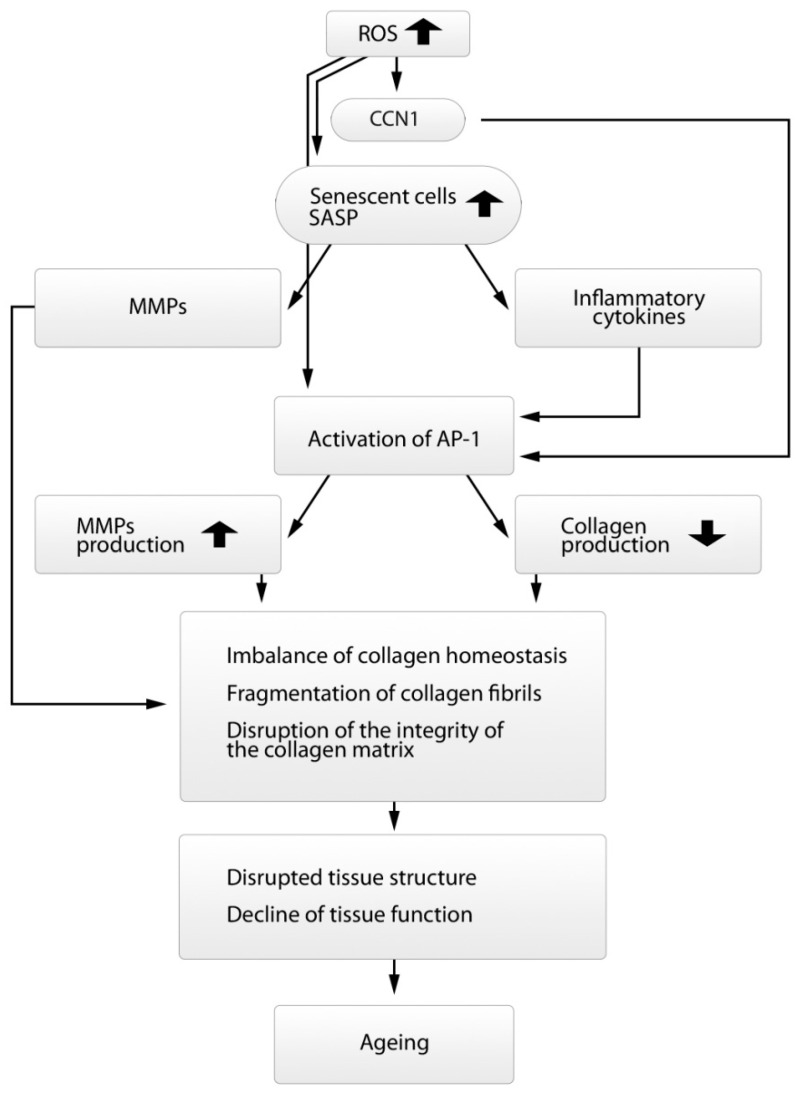
Scheme of the main signaling pathways involved in the process of collagen homeostasis disruption.

**Table 1 ijms-23-06655-t001:** Changes in glycosaminoglycans and proteoglycans in the skin with age.

	Chronological Aging	Photoaging
**HA in the** **epidermis**	↓ amount	↓ amount
**HA in the derma**	amount not changed ↓ bioavailability	↑ amount ↓ length
**Total content of sulphated GAGs**	↓ amount	↑ amount
**Versican**	↓ mRNA expression ↑ amount in males	↓ mRNA expression ↑ amount in the solar elastosis area
**Decorin**	↓ mRNA expression ↓ amount ↓ size	↓ amount in the solar elastosis area
**Biglycan**	↓ mRNA expression ↓ amount	↓ amount

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
