# Peer review of "Molecular Mechanisms of Changes in Homeostasis of the Dermal Extracellular Matrix: Both Involutional and Mediated by Ultraviolet Radiation"

_ijms, 2022, doi:10.3390/ijms23126655_

Round 1

Reviewer 1 Report

This is an interesting review dealing with skin aging and underlying dermis-epidermis interactions.  I have few comments, as follows.

Abstract and general comments :

I do agree that many skin ECM components are altered upon aging.  However, it is also known that, upon aging, dermal fibroblasts change their phenotype. With regard to several studies (notably those from J. Campisi’s lab but also many other labs), fibroblasts secrete inflammatory and ECM-degrading molecules. As well telomeres lenght is also altered upon aging. These topics are not discussed in this review. To me, these points should be adressed to complete the manuscript’s draft. Also, I would have appreciated some words about premature (skin) aging in some genetic diseases, including those linked to telomeres maintenance in some diseases.

Other points :

L 32-34, which photoaging ? it is not clear whether authors think of acute or chronic photoaging. Also, along the text, authors mention ultraviolet (UV) exposure. I would suggest to distinguish the different UV intervals since their respective effects differently affect the ECM, the dermal fibroblasts, and the overlaying compartment (epidermis) in human skin.

Figure 1, « Basament » membrane should be read basement.

Article organization could be improved. The general reader could face some difficulties to follow the order of presentation i.e. epidermis -> hypodermis -> dermis.

L 71, size of keratinocytes upon aging please quote the appropriate/original citation by Barrandon and Green, 1987.

L 75 & L 80 semi-desmosomes are better known as hemidesmosomes.

L 84, dealing with immunity, Langerhans cells are not the only cells to insure epidermal and dermal surveillance. Natural killer cells also diminish in numbers upon aging.

L 145, for the general reader, papillary and reticular dermis are not well defined.

L152, as mentioned above, the UV wavelenght(s) implicated in skin aging are not defined.

L 202-205, collagen 1 versus collagen 3 upon infammation and aging should be precised.

L 224-224, the half life of both collagen(s) and elastin shold be precised. « Long life » remains quite vague. Same for L 294 wich is quite misleading.

L 235, which MMPs are the authors talking about ?

L 320, AP1 is a complex of homo- hetero-dimeric complex ; I wonder which AP1 complex are talking about. This could be precised.

In conclusion, the review is interesting but, to me does not address all questions linked to aging in the field.

Author Response

We are grateful to the reviewer for a thorough analysis of our manuscript and tried to take into account all his comments. The point to point answers are below.

This is an interesting review dealing with skin aging and underlying dermis-epidermis interactions.  I have few comments, as follows.

Abstract and general comments:

I do agree that many skin ECM components are altered upon aging.  However, it is also known that, upon aging, dermal fibroblasts change their phenotype. With regard to several studies (notably those from J. Campisi’s lab but also many other labs), fibroblasts secrete inflammatory and ECM-degrading molecules. As well telomeres lenght is also altered upon aging. These topics are not discussed in this review. To me, these points should be adressed to complete the manuscript’s draft.

We have corrected: With age, there are also involutional changes in the fibroblastic differon of the dermis which are associated with the changes in amount of fibroblasts, their biosynthetic activity, and the formation of senescent fibroblasts that have a significant impact on the skin aging process (see review А. Zorina et al. (2022) [62].

Also, I would have appreciated some words about premature (skin) aging in some genetic diseases, including those linked to telomeres maintenance in some diseases.

This has already been described in detail by Wang A.S. and Dreesen, O. «Biomarkers of cellular senescence and skin aging. Front. Genet. 2018, 9, 247. in chapter GENETIC DISEASES AND SKIN AGING, therefore, we did not consider it necessary to describe this topic again.

Other points :

L 32-34, which photoaging ? it is not clear whether authors think of acute or chronic photoaging. Also, along the text, authors mention ultraviolet (UV) exposure. I would suggest to distinguish the different UV intervals since their respective effects differently affect the ECM, the dermal fibroblasts, and the  overlaying compartment (epidermis) in human skin.

We have corrected: chronic UV exposure, in particular, long-wavelength UVA, 320-400 nm [7]

Figure 1, « Basament » membrane should be read basement.

We have corrected  Figure 1:  «Basement» to “DEJ” (Dermo-epidermal junction)

Article organization could be improved. The general reader could face some difficulties to follow the order of presentation i.e. epidermis -> hypodermis -> dermis.

We have corrected the order -  epidermis -> hypodermis -> dermis  to epidermis -> dermis -> hypodermis.

L 71, size of keratinocytes upon aging please quote the appropriate/original citation by Barrandon and Green, 1987.

We have corrected: Barrandon, Y.; Green H. Three clonal types of keratinocyte with different capacities for multiplication. Proc Natl Acad Sci U S A  1987, 84, 2302-2306. doi: 10.1073/pnas.84.8.2302.

L 75 & L 80 semi-desmosomes are better known as hemidesmosomes.

We have corrected: hemidesmosomes

L 84, dealing with immunity, Langerhans cells are not the only cells to insure epidermal and dermal surveillance. Natural killer cells also diminish in numbers upon aging.

Unfortunately, we have not found any reliable data.

L 145, for the general reader, papillary and reticular dermis are not well defined.

We have corrected: The changes affect both layers of the dermis, the papillary is the upper and thinner layer, while the reticular  is the next more pronounced layer

L152, as mentioned above, the UV wavelenght(s) implicated in skin aging are not defined.

We have corrected: UVA long wavelengths

L 202-205, collagen 1 versus collagen 3 upon infammation and aging should be precised.

We have corrected: With age, the level of total collagen in the skin decreases (by approximately 1% throughout the entire adult person life [56]), and the level of main collagen types I and III also decreases, especially at the age of over 60 [57]. According to other data, the increase in collagen type III/I ratio occurs with age due to the increased degradation of type I collagen [58.]

L 224-224, the half life of both collagen(s) and elastin shold be precised. « Long life » remains quite vague. Same for L 294 wich is quite misleading.

We have corrected:  half-life of elastin and collagen in human tissues is 70  and 15 years, respectively

L 235, which MMPs are the authors talking about ?

We have corrected: MMPs comprise a family of endopeptidases that is responsible for physiological degradation of various extracellular matrix proteins, while MMP-1 (collagenase) is the main metalloprotease that begins the fragmentation of type I and III collagen fibers, after that the fibers are cleaved by MMP-3 (stromelysin) and MMP-9 (gelatinase) proteases

L 320, AP1 is a complex of homo- hetero-dimeric complex ; I wonder which AP1 complex are talking about. This could be precised.

We have corrected: AP-1 (c-Fos and c-Jun)

In conclusion, the review is interesting but, to me does not address all questions linked to aging in the field.

Reviewer 2 Report

The review "Molecular Mechanisms of Involutional Changes in Homeostasis of the Dermal Extracellular Matrix" of Zolina and co-workers gives an overview onto the effects of skin aging - UV and age mediated. This should also appear in the title. Otherwise the review is comprehensive and nicely written. I have some minor remarks:

Overall: Please try to cite primary literature and not reviews especially with Lee 2016 which is very close to your work.

l 90 so called glycosaminoglycans? It will be clearer to explain all structural players in the beginnings, like the involved GAGs and collagens.

l96 There are no DEJs shown in figure 1

l101-102 why don't you list these data in table 1 because it expands on the list of the cited publication

The abbreviation chop is not explained 

L129 which particular proinflammatory cytokines?

L139 what are fibrilar fibers?

In this context: in collagen biology it is distiguished between fibrils and fibers, please check your text accordingly that you use the right term in the right context

Explain elastosis

l228 don't use old fibers. These are typically damaged. In the context the MMP activity is part of tissue remodelling (which includes the degradation and replacement)

L230 use replaced by instead of restored through

L232 you mentioned 5 MMP groups - please elaborate

L234 the role of collagen type III is never explained (see my commentary to l90)

L266 integrin are not mentioned in this paper. Also not all integrins are able to bind collagens, and of them only two are known to bind collagen type I. Please be more specific

l300 the "the" is not needed

ROS play an important role in collagen cross-linking, this should be mentioned in the ROS part

Figure 4 Please revise this figure, it is confusing. It has aging as a conclusion at the top ant bottom of the figure, so any directionality is missing

 l338 The protein is called TGF-beta receptor type 2 and abbreviated TGFBR2 don't call it "type II receptor (TRII) to TGF-β"

l341 SMAD7 is no extra pathway but an inhibitory protein of the canonical TGF beta signalling pathway

L340/361/364 in this case replace expression with synthesis. Signalling pathways are working indirectly on the synthesis by modulation the expression. But modulation of the synthesis means more often change on the protein translation and posttranslational modification.

366 the small proteoglycan decorin

368 Decorin binds to cross-linked collagen fibers at the cleavage site for MMP-1,

393 This is not clear since you are proclaiming that glycation protects against MMP degradation

396 what do you mean with integrated buffer system, please specify in the introductory part

418 This are primary data directly taken from the cited publication. I am not sure if you can this and you don't have to present these data in particular.

422 This paragraph is already conclusion and not specific for the DF as the title suggests

442 use treatments instead of medicines

Author Response

We are grateful to the reviewer for a thorough analysis of our manuscript and tried to take into account all his comments. The point to point answers are below.

The review "Molecular Mechanisms of Involutional Changes in Homeostasis of the Dermal Extracellular Matrix" of Zolina and co-workers gives an overview onto the effects of skin aging - UV and age mediated. This should also appear in the title.

We have corrected: Molecular Mechanisms of Changes in Homeostasis of the Dermal Extracellular Matrix: both Involutional and Mediated by Ultraviolet Radiation

Otherwise the review is comprehensive and nicely written. I have some minor remarks:

Overall: Please try to cite primary literature and not reviews especially with Lee 2016 which is very close to your work.

We have corrected:  Meyer, L.J.; Stern R: Age-dependent changes of hyaluronan in human skin. J. Invest. Dermatol. 1994,102:385-389

Oh, J.H.; Kim, Y.K. Intrinsic aging- and photoagingdependent level changes of glycosaminoglycans and their correlation with water content in human skin. J. Dermatol. Sci. 2011, 62: 192-201.

l 90 so called glycosaminoglycans? It will be clearer to explain all structural players in the beginnings, like the involved GAGs and collagens.

 We have deleted «so called»

l96 There are no DEJs shown in figure 1

We have corrected figure 1: «Basement membrane» ->  «DEJ»

l101-102 why don't you list these data in table 1 because it expands on the list of the cited publication

Unfortunately, these data could not be included in the table since we did not find reliable indicators for photoaging.

The abbreviation chop is not explained

We have corrected: CHOP (C/EBP Homologous Protein, one of the transcription factors)

L129 which particular proinflammatory cytokines?

We have corrected: interleukin-6 (IL6), tumor necrosis factor α (TNFα), interferon γ (IFNγ)

L139 what are fibrilar fibers?

In this context: in collagen biology it is distiguished between fibrils and fibers, please check your text accordingly that you use the right term in the right context

We have corrected: The fibers (collagen, elastin, and fibrillar) ->  The fibers (collagen, elastin)

Explain elastosis

See in text L154-160 :“there is the deposition and accumulation of elastin masses possessing the incomplete molecular organization and, therefore, the incomplete function. This phenomenon is explained by the stimulating effect of UV on expression of the gene responsible for the synthesis of elastin. Solar elastosis zones are also characterized by the accumulation of proteoglycans [5, 10]”

l228 don't use old fibers. These are typically damaged. In the context the MMP activity is part of tissue remodelling (which includes the degradation and replacement)

We have deleted "old"

L230 use replaced by instead of restored through

We have corrected this

L232 you mentioned 5 MMP groups - please elaborate

We have corrected: MMPs comprise a family of endopeptidases that is responsible for physiological degradation of various extracellular matrix proteins, while MMP-1 (collagenase) is the main metalloprotease that begins the fragmentation of type I and III collagen fibers, after that the fibers are cleaved by MMP-3 (stromelysin) and MMP-9 (gelatinase) proteases

L234 the role of collagen type III is never explained (see my commentary to l90)

We have corrected: type I and III collagens are large fiber-forming collagens, the main structural components of ECM, type III collagen prevails.

L266 integrin are not mentioned in this paper. Also not all integrins are able to bind collagens, and of them only two are known to bind collagen type I. Please be more specific

We have corrected: It has been revealed that integrins (heterodimeric transmembrane proteins, specific receptors, primarily, α1β1 and α2β1 to collagen type I

l300 the "the" is not needed

We have corrected this

ROS play an important role in collagen cross-linking, this should  be mentioned in

the ROS part

We have corrected: One of the important factors in the formation of AGEs is oxidative stress [114]

Figure 4 Please revise this figure, it is confusing. It has aging as a conclusion at the top ant bottom of the figure, so any directionality is missing

We have corrected this in the Figure 4

 l338 The protein is called TGF-beta receptor type 2 and abbreviated TGFBR2 don't call it "type II receptor (TRII) to TGF-β"

We have corrected: TGFBR2 (TGF-beta receptor type 2)

l341 SMAD7 is no extra pathway but an inhibitory protein of the canonical TGF beta signalling pathway

We have corrected: Smad7, an inhibitory protein of the canonical TGF-β signaling pathway, which, in turn, blocks the TGF-β1 signaling pathway and suppresses the synthesis of procollagens

L340/361/364 in this case replace expression with synthesis. Signalling pathways are working indirectly on the synthesis by modulation the expression. But modulation of the synthesis means more often change on the protein translation and posttranslational modification.

We have corrected: As a result of an increase in the level of CCN1, collagen synthesis decreases and MMP-1 synthesis increases

366 the small proteoglycan decorin

We have corrected this

368 Decorin binds to cross-linked collagen fibers at the cleavage site for MMP-1,

We have corrected: Decorin binds to cross-linked collagen fibers at the cleavage site for MMP-1,

393 This is not clear since you are proclaiming that glycation protects against MMP degradation

We have removed this piece.

396 what do you mean with integrated buffer system, please specify in the introductory part

We have corrected: "the ground substance" (integrative buffer gel)

We removed this piece.

418 This are primary data directly taken from the cited publication. I am not sure if you can this and you don't have to present these data in particular.

We have removed this from the review.

422 This paragraph is already conclusion and not specific for the DF as the title suggests

We have removed this paragraph from the review.

442 use treatments instead of medicines

We have corrected “medicines for the treatments”